# *Trachemys scripta* Eggs as Part of a Potential In Vivo Model for Studying Sea Turtle Egg Fusariosis

**DOI:** 10.3390/jof11010023

**Published:** 2025-01-01

**Authors:** María Martínez-Ríos, Laura Martín-Torrijos, Javier Diéguez-Uribeondo

**Affiliations:** Department of Mycology, Real Jardín Botánico CSIC, Plaza Murillo 2, 28014 Madrid, Spain; maria.mr@rjb.csic.es (M.M.-R.); lmtorrijos@rjb.csic.es (L.M.-T.)

**Keywords:** pathogen, emerging fungal diseases, ascomycetes, pH, turtle, *Fusarium keratoplasticum*, multilocus sequence types (MLSTs)

## Abstract

The fungal pathogens *Fusarium keratoplasticum* and *Fusarium falciforme* are responsible for the emerging infectious disease named sea turtle egg fusariosis (STEF). This disease affects all sea turtle species throughout the world, causing low hatching success and mass mortalities. In this study, we investigated the potential use of widely available and easy-to-handle eggs of the invasive alien red-eared slider turtle, *Trachemys scripta*, as part of an in vivo host model to improve our knowledge of the biological properties of the pathogens responsible of the STEF. Specifically, we performed in vivo experiments, in which *T. scripta* eggs were challenged with conidia of *F. keratoplasticum* isolated from diseased sea turtle eggs. We found that the pathogen could colonize and develop similar signs to those observed in nature and fulfill Koch’s postulates. The pathogen showed high virulence properties (e.g., high disease incidence, severity, and low hatching success) and its ability to modify the pH in both the egg surface and culture media, confirming previously described fungal pathogen models. These results support the use of *T. scripta* as an experimental in vivo host model for studying the biological characteristics of STEF, thus providing valuable insights into the mechanisms underlying the emergence of this fungal disease.

## 1. Introduction

Over the past few decades, the incidence and virulence of fungal pathogens have increased along with an expansion in their geographic distribution and host range [1,2,3]. Diverse fungal taxa are responsible for emerging fungal diseases (EFDs) and for some of the worst declines in wildlife species, especially endangered animal species [1,2,4,5,6]. Some of the common key biological properties of emerging fungal pathogens allow them to manifest as EFDs [1,7]. For example, they (i) can show high virulence, i.e., the relative capacity of the pathogen to cause damage to the host [8]. This capacity is associated with rapid intra-host growth rates [9,10] and/or modulated pH changes in the host that allow for fungal colonization [11,12]. Previous studies have shown that certain ascomycete fungi, such as *Colletothrichum acutatum* [13], *Fusarium oxysporum* [14], and *Penicillium expansum* [15], can locally modify the host pH in response to carbon availability, thereby modulating the activation of genes involved in pathogenicity [11]. (ii) They possess long-lived environmental stages and the ability to survive as saprobes outside their host, facilitating their survival in diverse environments worldwide as cosmopolite organisms [1]. (iii) They are opportunistic (they depend on environmental or host conditions for infections) and generalist pathogens (they can infect a wide range of hosts), which allows the pathogen to easily disperse, infect, and generate high loads of inoculum. When pathogen thresholds for infection are reached and environmental conditions or immune system failure leads to the development of the disease, the pathogen manifests as an EFD [16].

The sea turtle egg fusariosis (STEF) is a recently described EFD [17] that affects sea turtle eggs worldwide. The disease is caused by the ascomycetous fungi *Fusarium keratoplasticum* and *Fusarium falciforme* [17,18], which belong to the *Fusarium solani* species complex (FSSC). These species have been described recently [19,20], and their identification requires a multilocus sequence type (MLST) approach based on the combined analysis of several loci [20]. These pathogens are globally distributed [21] and are associated with both low hatching success [18,22,23,24] and severe mass embryonic mortalities in the wild [18,22]. The characteristic signs of STEF in affected eggs include the presence of fungal mycelium growth on the egg surface and/or abnormal discolorations (e.g., yellowish, bluish, grayish, and reddish) on the eggshell, as described in Sarmiento-Ramírez et al. [18,22]. These pathogens appear to grow slowly on the egg surface and rapidly colonize internally when conditions are favored [18]. However, much remains unknown about their biological properties, infection processes, or virulence. Research on these aspects is challenging to conduct in situ due to adverse environmental conditions in tropical regions and the logistical difficulties of obtaining research permits for studies on threatened species [25]. Thus, identifying suitable model organisms is a significant advantage for this type of research. For example, the zebrafish has been a model to study the bacterial pathogenesis of *Streptococcus* spp. [26], *Pacifastacus leniusculus* for *Aphanomyces astaci* pathogenicity and invertebrate immunity [27], or *Galleria mellonella* for the fish pathogenicity of *Saprolegnia parasitica* [28].

Invasive alien species are ideal organisms for this purpose since they meet most of the requirements required for model organisms; i.e., they are widely available, easy to maintain and inexpensive, and assays can be easily replicated. For example, the red-eared slider turtle *Trachemys scripta* is a semiaquatic turtle listed among the World’s 100 Worst Invasive Alien Species [29] and as an invasive alien species of European Union concern [30]. During the last few decades, there are programs used for the control and eradication of this reptile in some European states [31,32,33], which make eggs of this invasive species easily available for experimental studies.

In this study, we assessed the utility of the eggs of the invasive species red-eared slider turtle, *T. scripta*, as a host model for in vivo experiments on the biological properties associated with the colonization of STEF-causing pathogens. For this purpose, we specifically (i) conducted an inoculation experiment in *T. scripta* eggs with *F. keratoplasticum* to investigate their pathogenic properties on the host model organism; (ii) calculated the disease incidence, disease severity, and hatching rate to evaluate the usefulness of the potential model organisms; (iii) checked the compliance of Koch’s postulates; and (iv) characterized the pH changes during fungal growth as one of the main biological properties of virulence.

## 2. Materials and Methods

### 2.1. Sample Collection

During the months of May to July (laying season of the species [34]) of 2019, a total of 564 *T. scripta* eggs from 94 nests were collected from El Marjal de Almenara wetland (Figure 1) in the province of Castellón (Spain) by the Department of Infrastructure, Territory and Environment of the Generalitat de Valencia as described in Martínez-Ríos et al. [21] and, subsequently, transported and maintained in an artificial incubator at 29.5 ± 0.5 °C for one day before starting the challenging experiments.

### 2.2. Inoculation of T. scripta Eggs with F. kertoplasticum

In order to replicate the STEF disease signs, we inoculated *T. scripta* eggs with a *F. keratoplasticum* isolate (RJB-FCR34A) obtained from a leatherback turtle (*Dermochelys coriacea*) at Pacuare Nature Reserve, Costa Rica, in 2018. We selected a total of 45 eggs from 30 nests without symptoms of STEF as described in Sarmiento-Ramirez et al. [18,22]. The eggs were randomly distributed in three plastic boxes (i.e., three experiment replicates) with sterile wet vermiculite as an incubating substrate. In these boxes, the eggs were placed in three rows with five eggs each (i.e., 15 eggs for each experimental replicate) (Figure 2). The first and second rows were used as controls. Eggs in the first row (control 1 hereafter) were not inoculated with the pathogen nor were they washed with distilled water before the experiment. Eggs in the second row (control 2 hereafter) were not inoculated with the pathogen, but they were washed with distilled water (pH = 7) to remove traces of clay from the surface before the start of the incubation period considering the fact that the spores of the fungus can persist in the soil for several years [35]. The third row (inoculated eggs hereafter) contained eggs that were first washed and then inoculated with a 6 mm diameter peptone glucose agar (PGA) disc with an actively growing fungal inoculum of *F. keratoplasticum*. The plastic boxes were incubated in two artificial incubators at 29.5 ± 0.5 °C, which correspond to the pivotal temperature of *T. scripta* eggs [36]. The eggs were checked daily during two months to detect fungal growth, and we sprayed sterile water (pH = 7) to keep the vermiculite moisture. During the entire incubation period, we macroscopically looked for the presence of common signs of STEF disease on eggs, i.e., the presence of fungal mycelium growth on the outside of the eggs and/or atypically colored areas (e.g., yellowish, bluish, grayish, reddish) on the eggshell as described in Sarmiento-Ramírez et al. [18,22].

### 2.3. Disease Incidence, Disease Severity, and Hatching Rate

During the two-month incubation period, we examined the presence of common signs of fungal infection on eggs. At the end of the incubation period, we assessed (i) the disease incidence, i.e., the number of infected eggs out of total over the incubation period [37]; and (ii) the disease severity, i.e., the level of the infection caused in the eggs, using a scale from 0 to 5 based on the percentage of egg shell surface affected, i.e., the infection coverage: 0 = no signs, 1 = ≤20%, 2 = 21 to 40%, 3 = 41 to 60%, 4 = 61 to 80% and 5 = >80% of the egg shell surface was diseased [38]; and (iii) the hatching rate as the number of eggs hatched out of the total number of eggs.

### 2.4. Reisolation of F. keratoplasticum from Diseased T. scripta Eggshell

In order to determine whether the infection signs were caused by the *F. keratoplasticum* isolate (RJB-FCR34A) used in the inoculation experiment, we verified that Koch’s postulates were fulfilled. Koch’s postulates are a set of criteria to identify whether a particular organism is the causative agent of a particular disease, based on the isolation of the pathogen from a diseased tissue, infection of a healthy host with the isolated pathogen, and the reisolation of the pathogen to prove causation of this infection process [39]. Thus, two months after the inoculation and coinciding with the period needed for the embryonic development for *T. scripta* [40], we selected from each inoculated egg a shell piece showing typical signs of STEF. In addition, we also took shell pieces from some of the control eggs to check the natural presence of *F. falciforme* and *F. keratoplasticum* in the eggs. From the selected areas of the eggs, we cut shell pieces (0.5 × 0.5 cm) and placed them onto PGA Petri dishes supplemented with ampicillin (100 mg/L) as described by Martínez-Ríos et al. [21,41]. The samples were incubated at 25 °C for 2–5 days until a mycelium was formed.

### 2.5. Molecular Identification of F. keratoplasticum Isolates from Experimentally-Infected T. scripta Egg Shell

Molecular characterization of the cultured fungal reisolates was performed by amplifying the nuclear DNA loci ITS nrDNA, LSU nrDNA, RPB2 nDNA, and TEF nDNA for each isolate, as used in the MLSTs approach [20] for identifying FSSC taxa, following the protocol described in Martínez-Ríos et al. [21]. A single round of PCR was performed according to Sarmiento-Ramírez et al. [18]. As a positive control, we used DNA extracted from an axenic culture of RJB-021FUS *F. keratoplasticum* from the Real Jardín Botánico-CSIC culture collection. Distilled MiliQ water was used as a negative control. Amplified products were visualized by electrophoresis in a 1% agarose gel stained with SYBR^®^ Safe DNA (Invitrogen^®^, Madrid, Spain). Positive amplified products were sequenced using an automated sequencer (MACROGEN, Inc., Madrid, Spain). The resulted sequences were visualized and edited with Geneious v.10.2.3 [42]. Finally, we conducted a BLAST search for each of the obtained sequences from each isolate to verify their identities in the GenBank database.

### 2.6. Physiological Characterization: pH Change

In order to check whether this fungal pathogen responsible for STEF produces pH changes according to the availability of carbon based on the fungal pathogen model described by [11], we measured the pH in a peptone glucose liquid media (PGl), which has high carbon concentrations, and in the inoculated egg shell surface, which has low free-carbon concentrations. For this purpose, we selected three isolates of *F. keratoplasticum* (RJB-FUS34A, RJB-FUS39A, and RJB-FUS37C) obtained from a leatherback turtle from Pacuare Nature Reserve. For measuring the pH in PGl, we transferred a 6 mm diameter agar disc with active fungal growth into 15 mL tubes containing 10 mL of PGl media for each isolate. We measured the pH of the medium for the following 7 days using a pH meter (micropH 2001, Crison^®,^ Fisher Manufacturer, Madrid, Spain). The pH values measured for each fungal pathogen species were plotted graphically, including the mean value of three replicates per fungal isolate, using the “ggplot2” package [43] in the software R v.3.4.1 [44].

### 2.7. Statistical Analysis

The values of the disease incidence, disease severity, and hatching rate from the selected treatments (control 1, control 2, and inoculated eggs) were statistically analyzed by a one-way ANOVA and subsequently compared with a post hoc Tukey test provided in the software R v.3.4.1 [44]. Furthermore, we compared whether there were significant differences between the treatments, i.e., inoculated eggs and controls using Student’s t-test in the software R v.3.4.1 [44].

The differences in pH values for each species were analyzed applying Student’s t-test in R v.3.4.1 [44]. For measuring pH values in egg shells, we used pH indicator strips (Merk KGaA, Darmstadt, Germany) on ten egg shells showing STEF signs and another ten egg shells free of STEF.

## 3. Results

### 3.1. Inoculation Experiment Using Eggs of T. scripta: Disease Development

At the time of collection, none of the eggs showed signs of STEF infection caused by *F. falciforme* and *F. keratoplasticum*, namely the presence of fungal mycelium and/or atypical colored spots on the egg shell as described below. After the first week following the inoculation, FSSC infection developed and the first signs of STEF disease appeared.

### 3.2. Disease Incidence, Disease Severity, and Hatching Rate in T. scripta Eggs

Differences in disease incidence between the experiment treatments (i.e., inoculated eggs, control 1, and control 2) were statistically significant (*p*-value < 0.001). A total of 14 eggs out of 15 inoculated with *F. keratoplasticum* showed development of *Fusarium* mycelium on the egg shell and/or typical colored spots of the STEF development (Figure 3). In control 1 (unwashed eggs), 5 of the 15 eggs showed signs of the infection, and in control 2 (washed eggs), 8 of the 15 eggs show signs of infection. However, there were no statistical differences between the replicates (*p*-value = 0.075) nor between both controls (*p*-value = 0.23). Therefore, results indicated that disease incidence in inoculated eggs was statistically higher than in control eggs (Table 1).

The results regarding disease severity were also significantly different (*p*-value < 0.001) between the experiment treatments (i.e., inoculated eggs, control 1, and control 2). The inoculated eggs showed differences in infection coverages: eight eggs had a 5-value in the scale (100% of the egg shell surface affected), four eggs had a 4-value (60–80% of the egg shell surface affected), and two eggs had a 1-value (less than 20% of the surface affected). The control eggs showed also different infection coverage. Thus, in control 1, nine eggs had a 0-value (no signs of infection); one egg had a 1-value (less than 20% of the egg shell surface affected); two eggs had a 2-value (20–40% affected), one egg had a 3-value (40–60% affected), and one egg had a 4-value (60–80% affected). In control 2, eight eggs had a 0-value (no signs of infection); one egg had a 1-value (less than 20% of the egg shell surface affected); two eggs had a 2-value (20–40% affected), three eggs had a 3-value (40–60% affected), and two eggs had a 4-value (60–80% affected). However, no controls were found with a 100% egg shell surface coverage by the pathogen. There were no significant differences between controls 1 and 2 (*p*-value = 0.75) nor between the replicates (*p*-value = 0.59). Therefore, results indicated that disease severity in inoculated eggs was statistically higher than in control eggs (Table 1).

There were no significant differences (*p*-value = 0.68) in hatching success between the experiment treatments (i.e., inoculated eggs, control 1, and control 2). Of the 45 eggs used in the experiment, only nine hatched (Figure 4): three eggs in inoculated eggs, two eggs in control 1, and four eggs in control 2 (Table 1). Seven of the nine hatching eggs had no signs of infection (*p*-value < 0.01), while the other two had less of 20% of their egg shell surface affected by infection.

### 3.3. Koch’s Postulates: Fungal Isolation and Molecular Characterization

We reisolated *Fusarium* colonies from all inoculated egg shells with signs of STEF infection and from two egg shells from control 1 and two from control 2, showing similar signs. BLAST search results for the four sequenced loci (i.e., ITS nrDNA, LSU nrDNA, RPB2 nDNA, and TEF nDNA) for the inoculated eggs showed that the isolates had a 100% BLAST identity to the *F. keratoplasticum* isolate used in the experiment (RJB-FCR34A) (GenBank accession number OL348272, OL348368, OL416122 and OL416123, respectively), except for some of them which showed a 100% identity to a *F. falciforme* sequence. In the controls, all the obtained sequences from the isolates had a 100% identity to *F. falciforme* sequences.

### 3.4. Physiological Characterization of F. keratoplasticum: Induced Changes in the pH

Colonies of *F. keratoplasticum* grown on PG1 (pH = 7) acidified the medium to a mean pH value 4.35 (sd ± 0.05) (Appendix A) after 7 days of incubation. When monitoring pH values in egg shells, we observed that the pH of all infected egg shell surfaces ranged from 8.0 to 9.0, while the pH of the non-infected egg shells ranged from 7.0 to 8.0.

## 4. Discussion

In this study, we demonstrated that eggs of invasive alien turtle species *T. scripta* can be used as an in vivo experimental host model to investigate key biological properties of the fungal pathogens responsible for STEF. We showed that STEF signs could be reproduced using this host model and confirmed the fulfillment of Koch’s postulates in the inoculation experiments. The pathogen exhibited high virulence properties with a high disease incidence, severity, low hatching success, and the ability to modify the pH of both the egg surface and culture media [1,11,12,45]. Therefore, these results confirmed that STEF pathogen *F. keratoplasticum* possesses key biological traits, such as high virulence, that enable it to manifest as an EFD [21].

However, we found that eggs used for the experiment were naturally exposed to *F. falciforme* either as a carrier or due to contact with spores in the soil. This explains why some control eggs showed signs of STEF, likely as a result of natural infections by *F. falciforme* in *T. scripta* eggs, similar to what occurred in the inoculated eggs. These fungal species are found worldwide, primarily as soil-associated species [18], and under suitable environmental conditions (moisture and temperature) [22], they can colonize the host [16]. In this study, the incubation conditions seem to have provided an optimal environment for the development of this naturally occurring fungal pathogen on the eggshells. As a result, the disease incidence and severity obtained in these infected control eggs were significantly lower than those of the inoculated. Although some control eggs were washed and cleaned to prevent these naturally occurring infections, this treatment was insufficient to fully prevent it. For future experiments using this model, we recommend collecting eggs from the turtles during the oviposition, preventing them from touching the soil as described by Sarmiento-Ramirez et al. [18]. 

The physiological characterization of *F. keratoplasticum* revealed evidence of advantageous biological properties that facilitate host colonization and virulence. We showed that the pathogen can induce pH changes in the host environment, which is a characteristic of several fungal pathogens [11,12]. Specifically, alkalinization of the egg shell (a substrate with poor carbon availability) was observed in infected areas, while acidification was detected in in vitro experiments under high carbon supply. This physiological behavior supports the pH modulation hypothesis proposed by Prusky et al. [11], which suggests that carbon availability in the environment is a key factor that triggers the secretion of small pH-modulating molecules, i.e., alkali under limited carbon or acids under excess carbon. These molecules activate some fungal genes encoding pathogenicity factors, depending on the environmental pH, thus enabling the pathogen to adapt to the host. Environmental alkalinization in fungi is commonly mediated by ammonia. This process has been described in the pathogens *Colletothricum acutatum* [13] and *Colletothricum gloeosporioides* [46], which causes anthracnose fruit rot; the rice blast fungus *Magnaporthe oryzae*, which causes high losses in rice crops worldwide [47]; or *Cryptococcus neoformans*, which infects humans and various animals [48]. In contrast, many other fungi acidify the environment to facilitate host tissue invasion. For example, *Pennicilium* sp. and *Aspergillus* sp. secrete gluconic and citric acids [15,49], while *Fusarium oxysporum* acidifies plant surfaces by producing fusaric acid [14]. Similar mechanisms may occur in *F. keratoplasticum* and *F. falciforme*, as these behaviors have been documented in other *Fusarium* species [11]. 

Further studies using *T. scripta* eggs as an experimental in vivo host-model will help unravel the molecular mechanisms underlying the infection process of these two pathogens. As a model organism, *T. scripta* offers several advantages over other turtle species: (i) it is widely distributed, (ii) easily available, and (iii) can be maintained under laboratory conditions. In Europe, this invasive species coexists and competes with the threatened native freshwater turtle, such as the European pond turtle (*Emys orbicularis*) and the Mediterranean pond turtle (*Mauremys leprosa*) [40,50,51,52,53]. One of the main purposes of conservation programs for these native species [54] is to control and eradicate the invasive red-eared slider turtle by removing gravid females and their eggs [55]. These conservation measures represent a source of *T. scripta* eggs, enabling a fast-replicative model system that is widely distributed, easily accessible and can be maintained under controlled conditions.

Finally, the use of this model system could enhance our understanding of STEF and facilitate the development of new treatments, preventive methods, and management protocols to address this disease.

## 5. Conclusions

*Trachemys scripta* eggs can be used as an experimental in vivo host model organism for studying the biological properties and the infection process of the fungal pathogens responsible for STEF disease.*Fusarium keratoplasticum* is capable of modulating the host environment pH to allow egg colonization.Using this model organism, we can now envision further studies focused on improving our understanding of factors involved in the pathogenicity of these fungal species and the environmental conditions conducive to sea turtle egg fusariosis development.

## Figures and Tables

**Figure 1 jof-11-00023-f001:**
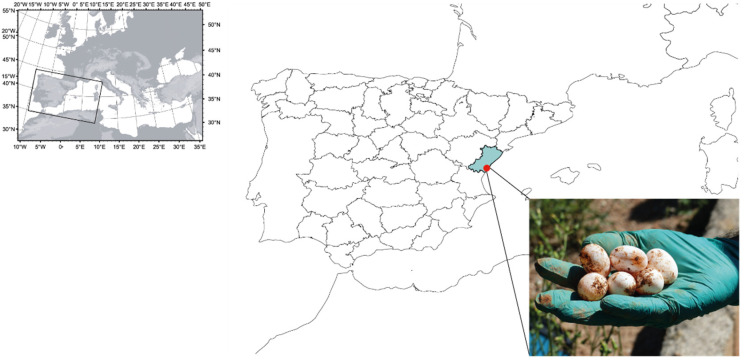
Sampling area of *Trachemys scripta* nests, El Marjal de Almenara wetland, Castellón (Spain).

**Figure 2 jof-11-00023-f002:**
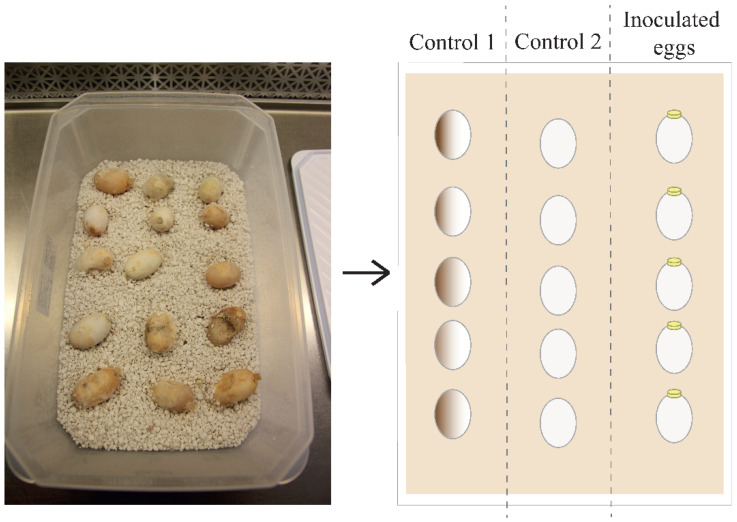
Design of the experimental in vivo host model using eggs of *Trachemys scripta*. Real (**left**) and schematic (**right**) image of an experiment carried out in plastic boxes with sterile wet vermiculite as an incubating substrate. In each experiment, 15 eggs arranged in three columns were used (five eggs/row). Each row represents one type of treatment (left to right): (i) control 1, which contains eggs as they came from the sample collection; (ii) control 2, which contains washed eggs with distilled water to remove traces of clay from the surface; and (iii) washed and inoculated eggs with growing fungal inoculum of *Fusarium keratoplasticum*.

**Figure 3 jof-11-00023-f003:**
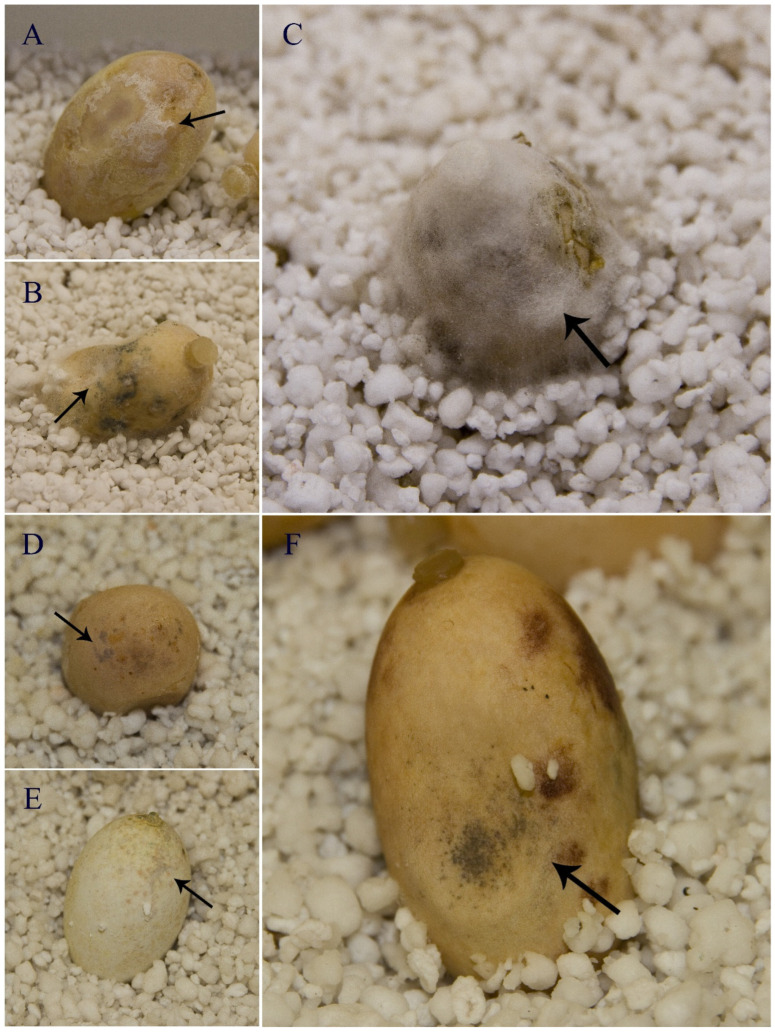
Signs of sea turtle egg fusariosis on eggs of *Trachemys scripta*. (**A**–**C**) Infected eggs showing white fungal mycelium growth around the egg shell (arrow) (scale bar = 1 cm). (**B**,**D**–**F**) Infected eggs showing the gray spots (arrow) typically associated with this disease (scale bar = 1 cm).

**Figure 4 jof-11-00023-f004:**
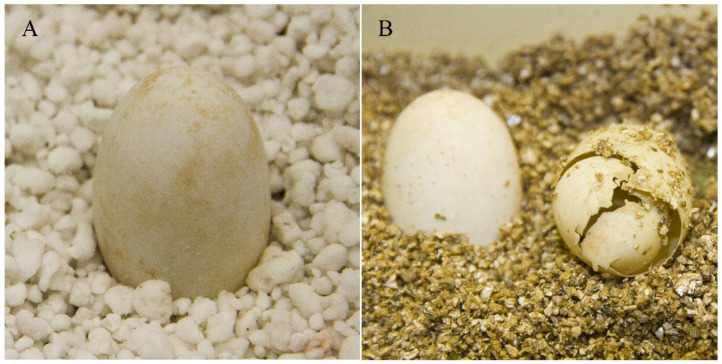
Non-infected eggs of *Trachemys scripta*. (**A**) An egg that shows no signs of fungal infection (scale bar = 1 cm). (**B**) Non-infected unhatched (left) and hatched (right) eggs (scale bar = 1 cm).

**Table 1 jof-11-00023-t001:** Disease incidence, disease severity, and hatching success of the in vivo experiment of *Trachemys scripta* eggs with *Fusarium keratoplasticum* (*Fk*).

Treatment	Disease Incidence (%) ^a^	Disease Severity ^b^	Hatching Success ^c^
Control 1 (non-treated eggs)	33.3	0.87 ± 1.3	2/15
Control 2 (washed eggs)	53.3	1.6 ± 1.55	4/15
Inoculated eggs with *Fk*	93.3 *	3.87 ± 1.73 *	3/15

^a^ Disease incidence is expressed as the percentage of the number of infected eggs out of the total eggs. ^b^ Disease severity is represented as a scale from 0 to 5 based on the percentage of egg shell surface affected by the infection with 0= no signs and 1 = ≤20, 2 = 21 to 40, 3 = 41 to 60, 4 = 61 to 80 and 5 = >80%. ^c^ Hatching success is represented as the number of hatched eggs out of the total eggs. * An asterisk denotes significant differences from other treatments at *p*-value > 0.05.

## Data Availability

All data supporting the findings of this study are presented within this article.

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
