# Peer review of "Trachemys scripta Eggs as Part of a Potential In Vivo Model for Studying Sea Turtle Egg Fusariosis"

_jof, 2025, doi:10.3390/jof11010023_

Round 1

Reviewer 1 Report

The introduction of the abstract is too long. In the abstract should be removed: As a consequence, there has been a slow improvement in our understanding of the crucial biological aspects of STEF.

In this paper, we investigate the potential use of widely available and easy-to-handle eggs of the invasive red-eared slider turtle, Trachemys scripta, as an in vivo host model to improve our knowledge of the biological properties of the pathogens responsible for STEF disease.

All keywords must be different from the title.
The introduction should be extended with more literature showing the importance of the work,

The isolate is from Costa Rica, it could be used in Europe. Comment on this in the discussion.
In figures 2,3 put a bar.
In the methodology, you used molecular tools. Although the n is small, it is acceptable.

It was demonstrated that eggs of invasive alien turtle species T. scripta can be used as an in vivo experimental host model to investigate key biological properties of the fungal pathogens responsible for STEF.

The article is well-written and deserves to be published after a few comments.

The introduction of the abstract is too long. In the abstract should be removed: As a consequence, there has been a slow improvement in our understanding of the crucial biological aspects of STEF.

In this paper, we investigate the potential use of widely available and easy-to-handle eggs of the invasive red-eared slider turtle, Trachemys scripta, as an in vivo host model to improve our knowledge of the biological properties of the pathogens responsible for STEF disease.

All keywords must be different from the title.
The introduction should be extended with more literature showing the importance of the work,

The isolate is from Costa Rica, it could be used in Europe. Comment on this in the discussion.
In figures 2,3 put a bar.
In the methodology, you used molecular tools. Although the n is small, it is acceptable.

It was demonstrated that eggs of invasive alien turtle species T. scripta can be used as an in vivo experimental host model to investigate key biological properties of the fungal pathogens responsible for STEF.

The article is well-written and deserves to be published after a few comments.

Author Response

Comments 1: The introduction of the abstract is too long. In the abstract should be removed: As a consequence, there has been a slow improvement in our understanding of the crucial biological aspects of STEF.
Response 1: the abstract was shortened as requested by the reviewer.

Comments 2: In this paper, we investigate the potential use of widely available and easy-to-handle eggs of the invasive red-eared slider turtle, Trachemys scripta, as an in vivo host model to improve our knowledge of the biological properties of the pathogens responsible for STEF disease.
Response 2: We have changed it according to the reviewer's suggestions.

Comments 3: All keywords must be different from the title.
Response 3: Keywords have been changed accordingly.

Comments 4: The introduction should be extended with more literature showing the importance of the work,
Response 4: We think we have already included the most relevant literature.

Comments 5: The isolate is from Costa Rica, it could be used in Europe. Comment on this in the discussion.
Response 5: The isolate in used was obtained with the required permits from the country and following the legislation of the recipient country.

Comments 6: In figures 2,3 put a bar.
Response 6: As the reviewer suggested, we have included the bar in the figures.

Comments 7: In the methodology, you used molecular tools. Although the n is small, it is acceptable.
Response 7: Yes, we agree with the reviewer and we consider that the n is acceptable for applying molecular tools.

Reviewer 2 Report

The manuscript is important in its area of study as it shows that T. scripta eggs can be used as in vivo model for the study of emerging fungal diseases of turtles

I  suggest the title be changed to “Trachemys scripta eggs as potential in vivo model for studying sea turtle egg  fusariosis”

The abstract should be revised to make it self-explanatory and standalone by reducing the background information and stating the experimental studies/methods in the study

51-55: There is need for the pathogenesis of STEF to be described. For example, is Fusarium a slow or rapidly growing mold? what is the incubation period from infection of turtle egg to exhibition of lesions on the egg? What are the symptoms seen on the eggs? This information is necessary to facilitate understanding of how uninfected eggs were selected in lines 99-100

73-73: There is need to highlight the conditions of incubation of the turtle egg. This information could answer why the eggs were washed

Check grammar errors in lines14,16,78,125,178,183,197,285,296,301,305: “to investigated” “easily handle”  “the potential used of” “Washed ages with distilled water” “following 7 days” “eggs shell” “show” “hypothesized proposed” “get infect host” “facilitate to unravel” “since is widely”

The second objective in lines 81-82 should be revised to be comprehensible

91: How long were the eggs kept before processing and what are the keeping conditions

106: This statement is not necessary “they were placed as they came from the sample collection”

What was the basis of washing the turtle eggs with water instead of disinfecting with alcohol which is volatile?

The authors should clarify why washed egg surfaces were not swabbed and cultured in peptone glucose broth and agar to ensure that spores of FSSC organisms did not colonize the egg surfaces prior to experimental infection. The statement in the discussion section, lines 275-276 could not be justified because of lack of culture of washed egg shells; and the recommendation of collecting eggs during oviposition would be solved if the egg shells are swabbed and cultured after washing/dinfection

111: Write the full meaning of PGA first time used

Statistical analysis should be a sub-heading under Materials and Methods. The authors erroneously mumbled up statistical analysis up with the experimental methodology in lines 138-141, 179-182.

The subheading 2.2 should be changed to “Inoculation of T. scripta eggs with F. kertoplasticum”

The subheading 2.4 should be changed to “Reisolation of F. keratoplasticum from diseased T. scripta eggshell”

Molecular identification of F. keratoplasticum isolates from experimentally-infected T. scripta egg shell should be a sub-heading

145-148: Narrated Koch’s postulates, therefore, is better suited for the Introduction section

153-155: The authors should clarify the following: Is it the petridishes that were supplemented with ampicillin (100mg/l) or the peptone glucose agar? Were the samples incubated or the inoculated agar plates? Moreover, mg/l should be corrected to mg/L, change ml to mL throughout the manuscript

182-184: It is not clear how pH indicator strip was used to measure the pH of egg shell

Table 1: Undefined abbreviation Fk and incorrect spelling “Inoculaed” should be corrected

Figure 4 described images G and H whereas the images were labelled A and B

240-249: Although the experiment focused on Fusarium, there is need for other molds that were reisolated from the eggshells to be mentioned. This could reveal the potential interaction of non-Fusarium species in STEF

302-304: Repetition of the statement in lines 256-258

306-314: Better suited in Introduction section for justification of why T. scripta eggs was investigated as a in vivo model for EFD

315-318: Already encompassed in the conclusion which should be written in a prose form

The authors should state the limitation(s) of the study

Author Response

Comment 1: I suggest the title be changed to “Trachemys scripta eggs as potential in vivo model for studying sea turtle egg fusariosis”
Response 1: We appreciate the suggestion of the reviewer and we changed the title as proposed.

Comment 2: The abstract should be revised to make it self-explanatory and standalone by reducing the background information and stating the experimental studies/methods in the study
Response 2: We have revised the abstract and changed it accordingly.

Comment 3: 51-55: There is a need for the pathogenesis of STEF to be described. For example, is Fusarium a slow or rapidly growing-mold? what is the incubation period from infection of turtle egg to exhibition of lesions on the egg? What are the symptoms seen on the eggs? This information is necessary to facilitate understanding of how uninfected eggs were selected in lines 99-100
Response 3: We have included in the text the required information requested.

Comment 4: 73-73: There is need to highlight the conditions of incubation of the turtle egg. This information could answer why the eggs were washed
Response 4: The reason why we washed the eggs was “to remove traces of clay from the surface before the start of the incubation period considering the fact that the spores of the fungus can persist in the soil for several years” as we explained in material and methods section (114-115).

Comment 5: Check grammar errors in lines14,16,78,125,178,183,197,285,296,301,305: “to investigated” “easily handle” “the potential used of” “Washed ages with distilled water” “following 7 days” “eggs shell” “show” “hypothesized proposed” “get infect host” “facilitate to unravel” “since is widely”
Response 5: We have corrected the grammar errors as suggested.

Comment 6: The second objective in lines 81-82 should be revised to be comprehensible
Response 6: We have rewritten the second objective as suggested to be comprehensible

Comment 7: 91: How long were the eggs kept before processing and what are the keeping conditions
Response 7: We have included the information suggested in the manuscript.

Comment 8: 106: This statement is not necessary “they were placed as they came from the sample collection”
Response 8: We have removed the sentence as proposed.

Comment 9: What was the basis of washing the turtle eggs with water instead of disinfecting with alcohol which is volatile?
Response 9: We decided “to remove traces of clay from the surface before the start of the incubation period because the spores of the fungus can persist in the soil for several years”, as we explained in the manuscript because the aim of doing that was to reduce the fungal load of spores.

Comment 10: The authors should clarify why washed egg surfaces were not swabbed and cultured in peptone glucose broth and agar to ensure that spores of FSSC organisms did not colonize the egg surfaces prior to experimental infection. The statement in the discussion section, lines 275-276 could not be justified because of lack of culture of washed egg shells; and the recommendation of collecting eggs during oviposition would be solved if the egg shells are swabbed and cultured after washing/dinfection
Response 10: We have rephrased the sentence to explain the questions of the reviewer. Briefly, we did not use ethanol because it can alter the shell composition and we did not use swabs because this method can not allow us to distinguish whether the pathogens are present chronically in the shell or saprophytically on the shell.

Comment 11: 111: Write the full meaning of PGA first time used
Response 11: We have changed the text as suggested.

Comment 12: Statistical analysis should be a sub-heading under Materials and Methods. The authors erroneously mumbled up statistical analysis up with the experimental methodology in lines 138-141, 179-182.
Response 12: We have modified the statistical analysis by including them in the sub-heading section “2.7. Statistical analysis” as suggested.

Comment 13: The subheading 2.2 should be changed to “Inoculation of T. scripta eggs with F. kertoplasticum”
Response 13: We have modified the subheading 2.2. as proposed.

Comment 14: The subheading 2.4 should be changed to “Reisolation of F. keratoplasticum from diseased T. scripta eggshell”
Response 14: We have modified the subheading 2.4. as proposed.

Comment 15: Molecular identification of F. keratoplasticum isolates from experimentally-infected T. scripta egg shell should be a sub-heading
Response 15: We have separated the molecular identification of F. keratoplasticum in the sub-heading section “2.5. Molecular identification of F. keratoplasticum isolates from experimentally-infected T. scripta egg shell” as suggested.

Comment 16: 145-148: Narrated Koch’s postulates, therefore, is better suited for the Introduction section
Response 16:
We have followed the reviewer's suggestion and still believe that this is best done in the Material and Methods section, where the process of the method is briefly described.

Comment 17: 153-155: The authors should clarify the following: Is it the petri dishes that were supplemented with ampicillin (100mg/l) or the peptone glucose agar? Were the samples incubated or the inoculated agar plates? Moreover, mg/l should be corrected to mg/L, change ml to mL throughout the manuscript
Response 17: The antibiotic is added to the culture medium, i.e. the PGA, and the plates containing the samples were incubated. We have corrected ml to mL as proposed.

Comment 18: 182-184: It is not clear how pH indicator strip was used to measure the pH of egg shell
Response 18: The pH strip is placed on the surface of the egg for a few seconds; it works by physical contact.

Comment 19: Table 1: Undefined abbreviation Fk and incorrect spelling “Inoculaed” should be corrected
Response 19: The abbreviation Fk is defined in the text of the table 1. We have corrected the incorrect grammar.

Comment 20: Figure 4 described images G and H whereas the images were labelled A and B
Response 20: We have corrected the text as suggested.

Comment 21: 240-249: Although the experiment focused on Fusarium, there is need for other molds that were reisolated from the eggshells to be mentioned. This could reveal the potential interaction of non-Fusarium species in STEF
Response 21: We think that this point is interesting but we also believe that it is out of the focus of this work and we agree that this model will allow to get further insides in interactions of other fungi with STEF.

Comment 22: 302-304: Repetition of the statement in lines 256-258
Response 22: We have removed the statement in lines 302-304.

Comment 23: 306-314: Better suited in Introduction section for justification of why T. scripta eggs was investigated as a in vivo model for EFD
Response 23: We think this is an option but we still believe that it fits better in the Discussion section.

Comment 24: 315-318: Already encompassed in the conclusion which should be written in a prose form
Response 24: We have corrected it accordingly.

Comment 25: The authors should state the limitation(s) of the study
Response 25: Limitations are explained in lines 265-278.